# Patterns of Dietary Fatty Acids and Fat Spreads in Relation to Blood Pressure, Lipids and Insulin Resistance in Young Adults: A Repeat Cross-Sectional Study

**DOI:** 10.3390/nu17050869

**Published:** 2025-02-28

**Authors:** Richard Woodman, Arduino A. Mangoni, Sarah Cohen-Woods, Trevor A. Mori, Lawrence Beilin, Karen Murphy, Jonathan Hodgson

**Affiliations:** 1College of Medicine and Public Health, Flinders University, Adelaide, SA 5001, Australia; arduino.mangoni@flinders.edu.au; 2Flinders Medical Centre, Adelaide SA 5001, Australia; 3College of Education, Psychology and Social Work, Flinders University, Adelaide, SA 5001, Australia; sarah.cohenwoods@flinders.edu.au; 4Flinders Institute for Mental Health and Wellbeing, Flinders University, Adelaide, SA 5001, Australia; 5Medical School, Royal Perth Hospital Unit, University of Western Australia, Perth 6000, Australia; trevor.mori@uwa.edu.au (T.A.M.); lawrie.beilin@emeriti.uwa.edu.au (L.B.); jonathan.hodgson@ecu.edu.au (J.H.); 6Clinical and Health Sciences, Alliance for Research in Exercise, Nutrition & Activity, University of South Australia, Adelaide, SA 5001, Australia; karen.murphy@unisa.edu.au; 7Nutrition and Health Innovation Research Institute, School of Medical and Health Sciences, Edith Cowan University, Joondalup 6919, Australia

**Keywords:** blood pressure, lipids, young healthy adults, K-means, Louvain, fatty acids, fat spreads

## Abstract

**Background/Objectives:** Determining whether dietary fatty acids and the use of fat spreads are associated with cardiovascular risk factors is difficult due to the multicollinearity of fatty acids and the consumption of multiple spread types. **Methods:** We applied clustering methodologies using data on 31 different fatty acids and 5 different types of fat spreads (high fat: butter, blended butters, and margarines; lower fat: polyunsaturated and monounsaturated) and investigated associations with blood pressure, serum lipid patterns and insulin resistance in the Raine Study Gen2 participants in Western Australia, at 20 and 22 years of age. **Results:** Amongst n = 785 participants, there were eight distinct clusters formed from the fatty acid data and ten distinct clusters formed from the fat spread data. Male participants had higher systolic blood pressure than females (122.2 ± 11.6 mmHg versus 111.7 ± 10.3, *p* < 0.001 at age 20 and 123.4 ± 10.6 versus 113.9 ± 9.8, *p* < 0.001 at age 22). Males consuming exclusively butter as a fat spread had significantly higher SBP (+4.3 mmHg) compared with males not using spreads. Males consuming a high intake of margarine had significantly higher SBP (+6.6 mmHg), higher DBP (+3.4 mmHg) and higher triglycerides (+30.5%). Amongst females, four patterns of fatty acid intake were associated with lower levels of HDL cholesterol compared with the low-saturated-fat/high *n*-3 reference group (*p* = 0.017 after adjustment for relevant confounders, range = −10.1% to −16.0%, *p* = 0.017). There were no associations between clusters and HOMA-IR or other serum lipids for males or females. **Conclusions:** Compared to using no fat spreads, amongst males, a high intake of margarine was characterised by higher systolic and diastolic blood pressure and higher serum triglycerides, whilst the use of butter also was associated with higher SBP. Diets low in *n*-3s or high in trans fats were associated with sub-optimal HDL levels amongst females.

## 1. Introduction

Dietary fats are well known to influence serum lipids and blood pressure [1,2]. A diet low in saturated fats, processed foods and *trans* fats and high in polyunsaturated fats including those found naturally in nuts, seeds and oily fish is now recommended for reducing the risk of cardiovascular disease (CVD) and other chronic conditions [3]. In addition to fats present in whole foods within the diet, the overall intake of dietary fatty acids is considerably influenced by the intake of added fats from butter, margarines, and other fat spreads [4]. Furthermore, highly processed spreads including some margarines, cooking fats for deep-frying and shortening for baking, contain trans-fatty acids that have adverse effects on cardiovascular health [5]. Monounsaturated and polyunsaturated fat spreads have become popular given the evidence that replacing saturated fatty acids with either monounsaturated or omega-6 (*n*-6) fatty acids lowers blood pressure [6]. These spreads may impact both blood pressure and lipids via both their fat content and fatty acid profile, and potentially via other food processing mechanisms that have recently been introduced in manufacturing fat spreads [7]. Debate therefore remains on which of the numerous forms of spread, if any, are the healthiest alternative in terms of their effects on cardiovascular risk factors [8].

Determining whether patterns of dietary fatty acid intake or the use of certain fat spreads are associated with detrimental changes in blood pressure and lipids independently is difficult for several reasons. First, fatty acids exist together in multiple food sources so that high intakes of one fatty acid result in high intakes of one or more other fatty acids [9,10]. In addition, many different fatty acids can be considered to have important impacts on nutritional status and health and therefore need to be accounted for in any analysis to accurately identify the associations of interest. However, their numerous and collinear nature means that commonly used statistical models are inadequate to determine independent associations between individual nutrients and health outcomes [11,12]. Specifically, adjusting for the intake of each individual fatty acid or of different food types becomes impractical or impossible due to the so-called “curse of dimensionality” which effectively restricts the number of variables that can be used in the adjustment process [13,14]. To overcome these problems, a common approach in dietary nutritional research is to focus on the patterns of nutrient intake, including the relative intake of fatty acids, and then to include these newly created patterns as labels in the analysis [15,16]. This approach is also a better option for examining exposures of interest, since foods or nutrients are generally not consumed in isolation within the diet [17]. Therefore, capturing patterns of real-world fat spread usage that incorporate fat spread combinations is a more realistic and practical approach to dealing with the potential for effect modification between spreads [18].

The K-nearest neighbors (K-NN) algorithm is an unsupervised learning method useful for clustering individuals by identifying each subject’s K-nearest neighbors according to a given set of variables of interest [19]. Community detection algorithms, used in network science, can then be applied to the K-NN network to further cluster individuals into distinct groups that exhibit similar profiles of nutrient intake. Such methods have previously been applied to detect clusters of cells and genes [20,21] and might be suitable for characterising patterns of fatty acid intake as well as patterns of spread usage. In this study, we aimed to first identify the patterns of fatty acid intake and fat spread intake within a representative sample of young adults that were part of a well-defined cohort of young adults from Western Australia. We then examined which, if any, of the fatty acid and spread patterns of intake were independently associated with clinic blood pressure, blood lipids and insulin resistance.

## 2. Materials and Methods

### 2.1. Participants and Study Design

The Raine Study (www.rainestudy.org.au, accessed on 1 December 2024) is owned by an Unincorporated Joint Venture (UJV), a collaborative partnership between the University of Western Australia (the centre agent), Curtin University, Edith Cowan University, Murdoch University, the University of Notre Dame, The Kids Research Institute Australia and the Women and Infants Research Foundation in Perth (WA) Australia. It represents a large cohort of Western Australian offspring studied from 18 weeks’ gestation to ascertain the relative contributions of familial risk factors, fetal growth, placental development, and environmental insults to outcome in infancy and to the precursors of adult morbidity. A total of 2900 women (Generation 1) were enrolled into this study from 1989 to 1991. There were 2868 live births, Generation 2 (Gen2), that have been prospectively followed up at regular intervals from birth, with demographic, lifestyle, clinical and biochemical information collected through questionnaires and clinical assessments [22]. The Human Ethics Committee at King Edward Memorial Hospital, Princess Margaret Hospital for Children, and the University of Western Australia in Perth approved all recalls of the cohort. The design of this study was a repeat cross-sectional study, that included only Gen2 participants with complete data for clinic blood pressure, blood lipids, glucose and insulin, and dietary fatty acids at both age 20 and 22. This study adhered to the Strengthening The Reporting of Observational studies in Epidemiology (STROBE) statement guidelines for enhancing the quality and transparency of health research (EQUATOR) [23]. The selection of these n = 785 participants from the full cohort is described in a Consolidated Reporting of Trials (CONSORT) flow chart (Appendix A). Informed, written consent was provided by them at each follow-up. A completed STROBE statement for cross-sectional studies is included as Appendix A.

### 2.2. Demographic and Clinical Data

The waist–hip ratio was measured at age 20 and 22 years from a clinical assessment. Computer-based follow-up health questionnaires relating to health and risk-taking behaviors including fruit and vegetable consumption, smoking and drinking was completed at age 20 and 22. Information on whether there was a family history of high blood pressure in either the mother or the father (Yes/No) was recorded at age 8 and was recorded as either “Yes” if either parent had a history and “No” if both parents had no history. Information on level of physical activity at age 20 was recorded as a 3-category variable (low/moderate/high) and as metabolic equivalents (METs) per week obtained from the number of minutes spent per day in moderate activity/vigorous activity/walking. The frequency of fruit consumption and frequency of vegetable consumption were recorded from an online questionnaire at age 17 using a 1–5 Likert scale with categories of rarely/1–2 times per month/1–2 times per week/3–5 times per week/6 + times per week. The standard number of alcoholic drinks consumed each day in the past week for each type of 20 different examples was obtained via the medical questionnaire administered at age 20. Responses were converted into grams alcohol per day and categorized as less than 1 standard drink, 1 to 3 standard drinks, or more than 3 standard drinks. Smoking status (yes/no) was obtained via the medical questionnaire completed at age 22. Additional information was obtained from Australia’s Commonwealth Scientific and Industrial Research Organization (CSIRO) semi-quantitative dietary recall questionnaire of food records [24,25] that was completed at age 17 including sodium intake (milligrams per day) and, standardized scores for a healthy diet and for a Western diet that were generated using a factor analysis of major food groups [26]. These two factor scores explained 13% (Western dietary pattern) and 8.5% (healthy dietary pattern) of the total variance in food intake.

### 2.3. Outcome Data

Clinic blood pressure was measured after an overnight fast with an appropriate cuff size for arm circumference, using a semi-automated oscillometric monitor (DINAMAP ProCare 100 vital signs monitor; GE Healthcare, Chicago, IL, USA). Recordings were obtained from the right arm after the participant had been seated for at least 5 min. Six measurements were taken 2 min apart and the first blood pressure recording was removed from the analysis [27]. The 5 separate readings for minutes 2 to 10 were then averaged. Trained phlebotomists obtained fasting blood samples prior to breakfast. The biochemistry assays for this study were conducted by PathWest Laboratories at Royal Perth Hospital. Serum triglycerides were measured using the Cobas MIRA analyser (Roche Diagnostics, Basel, Switzerland). Glucose was measured using an automated Technicon Axon Analyzer (Bayer Diagnostics, Sydney, Australia) and insulin was measured on an Immunlite 2000 Insulin Analyzer (Siemens Medical Solutions Diagnostic, Berkeley, CA, USA). Insulin resistance was determined using the homeostasis model assessment of insulin resistance (HOMA-IR), calculated as fasting plasma insulin (mU/L) × plasma glucose (mmol/L)/22.5 [28]. High-density lipoprotein (HDL) cholesterol was determined on a heparin–manganese supernatant [29]. Non-HDL cholesterol was calculated by subtracting HDL cholesterol from total cholesterol.

### 2.4. Exposure Data

Information on dietary intake at ages 20 and 22 was estimated using the 74-item semi-quantitative Dietary Questionnaire for Epidemiological Studies (DQESV2) Food Frequency Questionnaire (FFQ), developed by the Cancer Council of Victoria, Australia [30] and with validated good reliability compared to weighted food records [31]. The intake of 31 dietary fatty acid intake was estimated using the FoodWorks Version 2 Professional dietary analysis program (Xyris Software Pty Ltd., Brisbane, QLD, Australia) in conjunction with energy and nutrient data from the 2007 Australian Food and Nutrient Database (AUSNUT) [24,25]. The fatty acid common names, carbon-chain length and shorthand designation were: butyric acid (4:0), caproic acid (6:0), caprylic acid (8:0), capric acid (10:0), lauric acid (12:0), myristic acid (14:0), pentadecanoic acid (15:0), palmitic acid (16:0), margaric acid (17:0), stearic acid (18:0), arachidic acid (20:0), behenic acid (22:0), lignoceric acid (24:0), myristoleic acid (14:1*n*-7), pentadeconic acid (15:1), palmitoleic acid (16:1*n*-7), civetic acid 17:1*n*-9, oleic acid (18:cis1*n*-9) (accounts for 92% of monounsaturated dietary fatty acids), elaidic acid (18:1*n*-9 *trans*), gondoic acid (20:1*n*-9), erucic acid (22:1*n*-9), linoleic acid (18:2*n*-6), 18:2*n*-6 *trans*, α-linolenic acid (18:3*n-3*), dihomolinoleic acid (20:2*n*-6*)*, mead acid (20:3*n*-6), arachidonic acid (20:4*n*-6), eicosapentaenoic acid (20:5*n*-3), (22:4*n*-6), docospentaenoic acid (22:5*n*-3), and docosahexaenoic acid (22:6*n*-3) [32]. The DQESV2 FFQ [30] was also used to record usage (Yes/No) and intakes (grams/day) of spreadable fats. Participants were asked what type of spread they used on their bread, i.e., none/butter/butter-margarine blends/margarine/olive oil) and what type of margarine they used with examples, i.e., Canola/Sterol to lower cholesterol, e.g., Gold’n Canola, Meadow Lea Canola, olive oil margarine, e.g., Bertoli, Olive Grove, or Olivan or polyunsaturated margarine (no examples), “butter and margarine” blends (e.g., Devondale extra soft or Dairy Soft, Western Star spreadable varieties), and butter. By definition, butter, margarines and fat “blends” have a fat content of not less than 80%, whereas ‘fat spreads’ and ‘fat blend spreads’ have less than 80% fat [33]. The variables in the dataset had the following Victorian Cancer Council survey labels; None, Marg, Polunsaturated_marg, Monounsaturated_marg, Marg_butter blends, and Butter.

### 2.5. Clustering of Dietary Fatty Acids and Fat Spreads

Patterns of fat spread usage and dietary fatty acid intake were identified by using K-nearest neighbors (K-NN) and Louvain community detection algorithms to create subject clusters (see Appendix A for full details) that were later used as the exposure variables of interest in regression analysis. The raw data used for creating the fat spread clusters were the mean intake (grams per day) across ages 20 and 22 for each of 5 types of spread recorded using the DQESV2 Cancer Council questionnaire (5 variables). Similarly, the data used for fatty acid clustering were the mean intake of each of 31 dietary fatty acids estimated at ages 20 and 22 using the FoodWorks Version 2 dietary analysis program (31 variables in total). Briefly, the clustering approach involved 3 steps which used (1) a K-nearest neighbors (K-NN) algorithm, (2) generation of a K-NN network and (3) community detection. The method has previously been applied to detect clusters of cells and genes [20,21]. For this study, the extracted dietary information on the estimated mean dietary intake of 31 fatty acids estimated at ages 20 and 22 was used with the K-NN algorithm to cluster participants such that those with more similar fatty acid intakes were more likely to be one of the 20 nearest neighbors of one another in a participant-participant matrix. The cosine similarity method was then used to transform the K-NN distances of the 20 nearest neighbors into edge weights for a K-NN network. Finally, the Louvain community detection algorithm was applied to detect the clusters of participants in the network based on their fatty acid intake profiles. The same method was repeated separately for the data on the 5 fat spread types recorded at ages 20 and 22 (12 variables in total including “no spread”).

### 2.6. Statistical Analysis

The study population was described using the mean and standard deviation for normally distributed variables, the median and inter-quartile range for non-normally distributed variables, and frequencies and percentages for categorical variables. Differences between males and females were assessed using an independent *t* test or Mann–Whitney U test for continuous variables, and a chi-squared test for frequency data. The mean intakes (grams per day) of fatty acid spreads (None, Margarine (Yes/No), Polyunsaturated margarine, Monounsaturated margarine, Margarine–Butter Blends, Butter) were described using bar plots with 95% confidence intervals. The mean intake of fatty acids was described using line plots of the standardized means for each fatty acid cluster.

Generalized linear models were used to assess whether there were differences in blood pressure, lipids, and insulin resistance between fat spread clusters. Since several of the outcomes of interest showed significant interaction effects for males and females, analysis was performed separately for each. The exposures of interest for each model were the spread cluster and the fatty acid cluster which were included as categorical variables. Two separate models were used for the analysis. Model 1 did not adjust for potential confounders. Model 2 included adjustment for age, the waist–hip ratio, family history of hypertension, Healthy and Western diet scores, frequency of fruit and vegetable consumption, alcohol intake, sodium intake, METs per week, and physical activity category (0/1/2). Robust standard errors were used to account for the clustering of the data for each subject (at visits aged 20 and age 22). To control for multiple comparisons between the reference cluster and other clusters, a familywise 2-sided type 1 error rate of alpha = 0.05 was used and only if significant (*p* < 0.05), were individual differences then also assessed. Coefficient plots with 95% confidence interval of the mean difference (β) or geometric mean ratio were generated. The reference cluster for the spread cluster variable was “No spreads”, and the reference cluster for the fatty acids cluster variable was “Low-saturated-fat/high- omega-3 (*n*-3)”. Due to missing data for some of the covariates (BMI, family history of hypertension, healthy diet score, Western diet score, alcohol consumption, smoking status, sodium intake, fibre, fruit intake, vegetable intake, alcohol intake, METs per week, and physical activity category (0/1/2), multiple imputation with chained equations was used to create 20 multiply imputed datasets which were used for all models (see Appendix A for further details of missing data). Predictive mean matching was used to impute family history of hypertension, smoking category and physical activity category, and linear regression was used for imputing the other covariates. Non-missing variables included in the imputation were systolic and diastolic blood pressure measurements, and indicators for subject ID, and visit (at aged 20 versus 22). All analyses were performed using Stata (version 17.0, StataCorp, College Station, TX, USA).

## 3. Results

### 3.1. Study Population

From the original 2868 Generation-2 participants, there were 785 participants with complete data for dietary fatty acids, fat spreads, clinic blood pressure and serum lipids, glucose and insulin, recorded at both age 20 and at age 22 (see Appendix A for CONSORT flow diagram). Table 1 describes the demographic and clinical characteristics of the participants including their blood pressure and information on diet.

There were slightly more males (52.1%) than females (47.9%), and the overall BMI across all participants was 24.2 ± 4.7 kg/m^2^ at the visit aged 20 and 25.0 ± 5.0 kg/m^2^ at the visit aged 22. The overall mean systolic blood pressure (SBP, 5 measures per participant) was 122.2 ± 11.6 and 111.7 ± 10.3 mmHg for males and females, respectively (*p* < 0.001), at age 20 and 123.4 ± 10.6 and 113.9 ± 9.8 mmHg for males and females, respectively (*p* < 0.001), at age 22. Total cholesterol was slightly higher for females versus males at age 20 (4.52 ± 0.80 versus 4.16 ± 0.74 mmol/L, *p* < 0.001) and at age 22 (4.71 ± 0.81 versus 4.54 ± 0.79 mmol/L, *p* = 0.003). However, HDL cholesterol was higher and non-HDL cholesterol was higher for females versus males at age 20 (*p* = 0.010). Western diet factor scores and similar healthy diet factor scores were similar to the Raine Study Gen2 participant full cohort [26] with means (±SD) of −0.06 ± 0.82 and 0.07 ± 0.90, respectively, and participants were also generally meeting fruit, vegetable and alcohol recommendations at age 20 and 22 according to the Australian Guide to Healthy Eating (AGHE) [35]. Females had a slightly healthier profile than males for the Western diet score (−0.36 ± 0.68 versus 0.26 ± 0.83, *p* < 0.001) but were similar for the healthy diet score (0.13 ± 0.89 versus 0.01 ± 0.92, *p* = 0.163). Median (IQR) alcohol consumption (at age 20) was similar between males [10 (0–27.1) grams/day] and females [5.7 (0–15.7)] grams/day, *p* = 0.132.

### 3.2. Dietary Fat and Fat Spread Clusters

Appendix A displays the eight dietary fatty acid clusters detected amongst the 785 participants using data for the mean of intake of 31 dietary fatty acids at ages 20 and 22 and applying the K-NN clustering and Louvain community detection algorithms. The clusters were well separated (modularity = 0.664) and ranged in size from n = 44 (cluster 7) to n = 156 (cluster 3). There were significant differences in fatty acid intakes across clusters for all fatty acids (*p* < 0.001 for each) (Figure 1). The fatty acid clusters were subjectively labelled as FA0: Low-Sat/High16:1, FA1: ModSat/Low16:1, FA2: Mod16:1/ModN-3, FA3: High-Sat/Low N-3, FA4: Mod-Sat/LowTrans&N-3. FA5: Low-Sat/High18:1T, FA6: Low-Sat High18:1T&18:2N6, and FA7: Low-Sat High N-3.

Appendix A displays the 10 dietary fat spread clusters detected amongst the 785 participants using data for the mean intake of 5 different fat spreads at ages 20 and 22 and applying the same K-NN and Louvain algorithms. There was a high degree of clustering (modularity = 0.834) with clusters ranging in size from 31 (cluster 6) to 163 (cluster 3). The 10 spread clusters were subjectively labelled as S0: None, S1: Mod Marg/Blend, S2: Butter, S3: Mod Marg, S4: High Blends, S5: High Marg, S6: Marg/Poly, S7: Poly/Mono, S8: Marg/Butter, and S9: Butt/Blends.

There were significant differences in fat spread intakes across clusters for all fat spreads (*p* < 0.001 for each) (Figure 2).

There was a significant association between the fatty acid clusters and the fat spread clusters (χ^2^ = 427.0, 63 df, *p* < 0.001) (Appendix A).

### 3.3. Association Between Fat Spread Clusters, Fatty Acid Clusters, and Blood Pressure, Lipids and HOMA-IR

Regression analysis for blood pressure, serum lipids and HOMA-IR on the fatty acid clusters and fat spread clusters revealed associations between fat spread cluster membership and systolic BP (SBP), diastolic BP (DBP), and serum triglycerides. The significant associations are summarized in Table 2 and illustrated in Figure 3 (SBP), Figure 4 (DBP) and Figure 5 (triglycerides) and the non-significant associations are illustrated in Appendix A (clinic pulse pressure (PP)), Appendix A (HOMA-IR), Appendix A (Cholesterol), Appendix A (HDL cholesterol) and Appendix A (Non-HDL cholesterol). The coefficients in these figures are shown for models with and without adjustment and for males and females separately. In unadjusted analysis there was an overall difference in SBP across spread clusters for males (*p* < 0.001) which remained after adjustment (*p* = 0.010). Compared to males consuming no spreads (cluster 0), SBP was higher for males in spread cluster 2 (Butter) (β = 4.26 mmHg, *p* = 0.007), and cluster 5 (high margarine) (β = 6.61 mmHg, *p* = 0.001). There was an overall difference in DBP across fat spread clusters for males after adjustment (*p* = 0.035) with higher DBP in males in the high margarine cluster (β = 3.4 mmHg, *p* = 0.012). Higher triglycerides were observed in four of the nine clusters that used spreads for males (*p* = 0.003 across clusters) including high margarine (+26.4%, *p* = 0.009), margarine/polyunsaturated fats (+30.5%, *p* = 0.005), polyunsaturated/mono (+22.1%, *p* = 0.004), and margarine/butter (+29.1%, *p* < 0.001). Amongst females, in the adjusted model (but not in the unadjusted model), there were significantly lower levels of HDL cholesterol for four of the fatty acid patterns versus the reference cluster (low saturated fats and high *n*-3s) (*p* = 0.017) (Appendix A). This included fatty acid cluster 1 (Moderate Saturated/Low 16:1) (−12.4%, *p* = 0.033), cluster 2 (Moderate 16:1/Moderate N-3) (−10.1%, *p* = 0.046), cluster 3 (High saturated/low N-3) (−14.0%, *p* = 0.055), cluster 4 (Moderate saturated/Low Trans and N-3) (−15.0%, *p* = 0.003) and cluster 5 (Low saturated/high 18:1T) (−16.0%, *p* = 0.003). There were no significant differences between cluster membership and either clinic pulse pressure, HOMA-IR, total cholesterol, and non-HDL cholesterol (Appendix A) for either males or for females.

## 4. Discussion

In this study, we used a combined K-NN algorithm and Louvain community detection algorithm approach to objectively cluster young healthy adults based on both their patterns of averaged dietary fatty acid intake and on their fat spread usage. The input data were obtained from averaging dietary questionnaire and calculated nutrient data obtained at ages 20 and 22. Based on the averaged intake of 31 different fatty acids, there were eight different patterns identified, and based on the averaged intake of five different types of spread, there were ten patterns identified. When included together within a regression model, the results provide evidence for significantly higher clinic systolic blood pressure amongst males consuming either butter alone or high intakes of margarine. There was also evidence for higher diastolic blood pressure and higher triglycerides amongst males consuming high intakes of margarine, and higher triglycerides amongst males consuming polyunsaturated or monounsaturated spreads. These effects persisted after adjusting for confounders. Amongst females, there were also statistically significant lower HDL cholesterol levels amongst several clusters of females whose pattern of dietary fats was different to the low-saturated-fat/high *n*-3 dietary pattern used as the reference cluster. However, these effects were relatively small with sizes between 10 and 15 percent lower.

The two different patterns of fat spread usage that were independently associated with higher clinic SBP amongst males were butter alone and a high intake of margarine (4.3 mmHg and 6.6 mmHg, respectively). Serum triglycerides were also 26.4% higher in males with high intakes of margarine. These differences in SBP are clinically meaningful, even amongst this population of young adults [37], of whom a small proportion demonstrated high normal or stage 1 hypertension. The reasons for the observed associations in this study cannot be determined from this study itself but include well-documented harmful effects of fat spreads on lipid profiles, the detrimental effects of high saturated fat diets on insulin resistance and the metabolic syndrome, a relationship between the identified clusters and the preference for high-fat foods, or the harmful effects of trans-fatty acids on serum lipids including LDL cholesterol and HDL cholesterol.

Butter, high saturated fat margarines, and polyunsaturated spreads have all been associated with worse lipid profiles in comparison to margarines high in monounsaturated fats [38]. Energy dense diets are also strongly correlated with the development of obesity and insulin resistance, symptoms associated with metabolic syndrome [39] and excess consumption of saturated fat in particular is strongly associated with insulin resistance [40]. The metabolic syndrome that consists of insulin resistance, higher blood pressure, and abnormal cholesterol or triglyceride levels [41] is also associated with arterial stiffening [42] which precedes hypertension in young adults [43]. Alternatively, our results might reflect previously reported associations between taste phenotypes and CVD risk [44] since the clusters that had worse blood pressure and lipids were characterised by a higher consumption of either margarines or butter which by definition contain 80% + fat content. However, additionally adjusting for the percentage of energy in the diet derived from fat did not change any of the results. Our findings may also partly reflect the well-documented negative effects of consuming trans-fatty acids on LDL and HDL cholesterol [45] given that Australia remains one of the few high income countries, alongside New Zealand, Japan, and South Korea that have not taken effective action on banning trans-fats [46]. However, despite this, most major food processing companies in Australia have changed their processing methods to reduce trans fats in their products [47], and they largely disappeared after 1996 [48]. Industrially produced trans fats in other Australian foods are also now amongst the lowest in the world [49]. The participants in this study were aged 22 between July 2011 and July 2014 and it is therefore unlikely, that the increases in blood pressure and triglycerides amongst consumers of butter and margarine were due to higher intakes of trans-fatty acids from these food sources. Finally, although we also adjusted for Western and healthy diet factor scores in our analysis, there remains the potential for residual confounding due to self-reporting of diet and the nature of the data obtained from the DQESV2 FFQ that relies on memory of dietary habits for the previous 12 months. We also did not account for the different sources of the fatty acids obtained from the diet including types of fish or the intake of *n*-3 supplements. However, the impact of the latter would likely be small given that only 28 of 491 participants (5.69%) reported using *n*-3 supplements at age 17.

Our study identified some notable differences in the observed dietary cluster–risk factor associations for the male and female participants, with cluster–blood pressure and cluster–triglyceride associations identified for males but not females and cluster–HDL cholesterol associations identified for females but not males. One explanation for these different findings might be the different baseline levels for each risk factor between males and females. Males had an approximate 10 mmHg higher systolic blood pressure than females at both age 20 and age 22. Similarly, HDL cholesterol levels were approximately 15 to 20 percent higher in females than in males and were more than 40 percent higher in their variability (standard deviations). Triglyceride levels were similar for both males and females at each age, but there were no significant cluster–triglyceride associations for females. However, there was a distinctly similar trend in the observed cluster–triglyceride associations within females to those observed for males.

Our study had several strengths. Firstly, it applied a clustering approach to overcome the problems associated with multicollinearity, high dimensionality, and the many interactions between nutrients. By considering the numerous fatty acids as a more limited set of dietary patterns, we could examine whether patterns of spread usage or patterns of fatty acid intake are independently associated with blood pressure and lipids in young healthy adults. In addition to novel findings that demonstrate an association between patterns of fat spread usage, blood pressure and triglycerides, our data also confirmed that the most favourable pattern of fatty acid intake for lower blood pressure and lipids is that of the Mediterranean diet that is low in saturated fats and trans-fatty acids and high in *n*-3 fats [50]. Although replacement of butter with unsaturated fats is known to improve lipid profiles [51], consumption of any type of fat generally leads to worse lipid profiles [52].

Our study had several limitations. Although we adjusted for many potential confounders, some of the dietary information used in the adjustment process was obtained at age 17 rather than at ages 20 and 22 when the blood pressure data were obtained. This included the information on Healthy and Western diet scores, fruit, vegetable, alcohol, fibre, and sodium intake. However, the diets of the Raine Study Gen2 participants are known to track closely into adulthood [24] and therefore adjusting for these measures should still reduce rather than increase the potential for bias. Other sources of bias include the measurement of sodium which was estimated using a semi-quantitative food questionnaire and may not be a reliable indicator of salt intake compared to using spot urines which were unavailable. There may have been selection bias of healthier participants since the number of participants with complete measures used for this study were several hundred fewer than the total number of subjects that responded at ages 20 and 22 to the surveys. Secondly, although we established an association between different patterns of spread usage, blood pressure and lipids, due to the observational nature of our study we cannot assign causality to our findings. It is possible that the patterns of intake may simply be markers for some other aspect of lifestyle behaviour such as levels of physical activity, although we reduced the potential for this by adjusting for the waist–hip ratio, levels of physical activity and metabolic equivalents per day. Similarly, although patterns of spread intake may also be a marker of the intake of other processed foods, we adjusted for both Western diet scores and healthy diet scores previously obtained from factor analyses. It is also possible that the observed associations may be due to reverse-causality. For example, participants that had knowledge of having a higher blood pressure or lower HDL cholesterol might influence them to alter their diet to a healthier diet. Participants with normal values may also be tempted to follow a less healthy diet. Such a potential for reverse-causality is however reduced given that the information on both diet and risk factors was recorded within a tight timeframe for each recall of the cohort, and at two separate recalls separated by two years, making the potential for the observed associations to be driven by risk factor knowledge and subsequent dietary changes less likely. The potential for reverse-causality is also less likely given that the direction of the observed associations confirmed previously established causal harmful associations between less healthy diets and higher levels of cardiovascular risk factors.

### Implications for Clinical Practice and Directions for Future Research

Overall, our study highlights the risks with high intakes of margarine or butter and the benefits of a low-saturated-fat/high *n*-3 diet in a population of young adults suggesting that adherence to a diet with this fatty acid profile should be adopted as early as possible. Our data also suggest that it would be valuable to consider how the growing role of Artificial Intelligence (AI) in nutrition and health research that includes the analysis of dietary patterns such as here and elsewhere [53] might be best utilized to assist in the management of cardiovascular risk factors [54]. The integration of AI-based technologies within healthcare settings has the potential to significantly improve nutritional analyses and contribute to the development of greater interdisciplinary care that combines nutrition, nursing, clinicians, and technology [55]. These advancements could lead to more personalized dietary recommendations [55] and improved monitoring of diet-related health outcomes over time.

## 5. Conclusions

We have shown significant associations between patterns of dietary fatty acid intake and patterns of fat spread usage with SBP and serum triglycerides in young healthy males and females. Patterns of fat spread intake that included high-fat spreads (margarine or butter) were, amongst males, associated with higher blood pressure and/or lipids than male participants that used no spreads at all. Females consuming diets that were either low in *n*-3, higher in trans-fatty acids, or higher in saturated fats had lower HDL cholesterol than those consuming a low-saturated-fat/high *n*-3 diet. Pending further research investigating cause–effect relationships, our results highlight the risks with high intakes of margarine or butter and the benefits of a low-saturated-fat/high *n*-3 diet.

## Figures and Tables

**Figure 1 nutrients-17-00869-f001:**
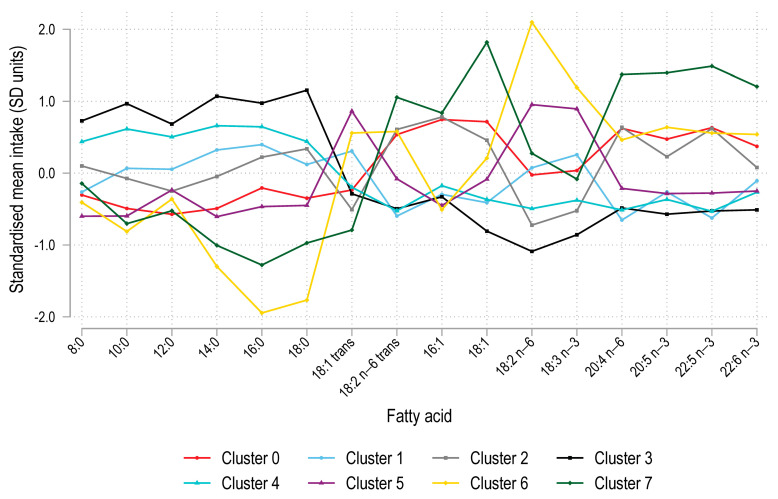
Fatty acid intakes across clusters for the 16 major fatty acids [36]. Figure legend: 8:0 = caprylic acid, 10:0 = capric acid, 12:0 = lauric acid, 14:0 = myristic acid, 16:0 = palmitic acid, 18:0 = stearic acid, 16:1*n*-7 = palmitoleic acid, 18:1 = oleic acid, 18:1 trans = elaidic acid, 18:2 *n*-6 = linoleic acid, 18: 2*n*-6 trans= linoelaidic acid, 18:3*n*-3 = α-linolenic acid, 20:4*n*-6 = arachidonic acid, 20:5*n*-3 = eicosapentaenoic acid, 22:5*n*-3 = docospentaenoic acid, and 22:6*n*-3 = docosahexaenoic acid.

**Figure 2 nutrients-17-00869-f002:**
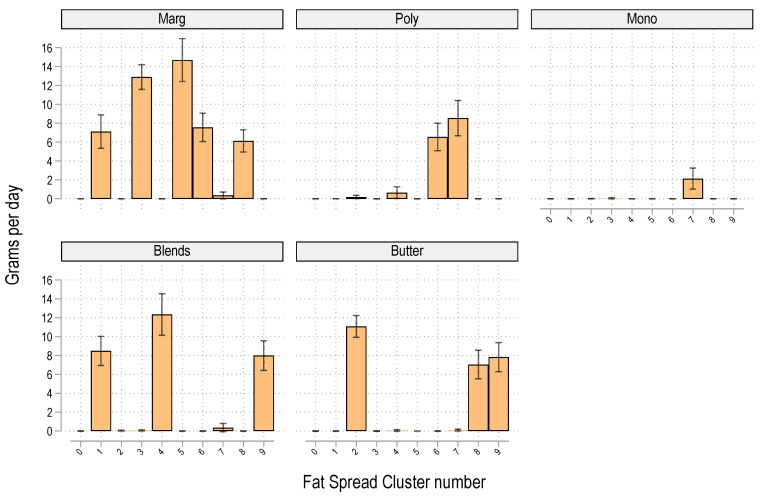
Mean fat spread usage by cluster. Marg = margarine, Poly = polyunsaturated margarine, Mono = monounsaturated margarine (high in either canola oil, e.g., Gold’n Canola, Meadow Lea Canola, or olive oil, e.g., Bertoli, Olive Grove, or Olivan), and blends = Margarine–Butter blends (e.g., Devondale extra soft or Dairy Soft, Western Star spreadable varieties).

**Figure 3 nutrients-17-00869-f003:**
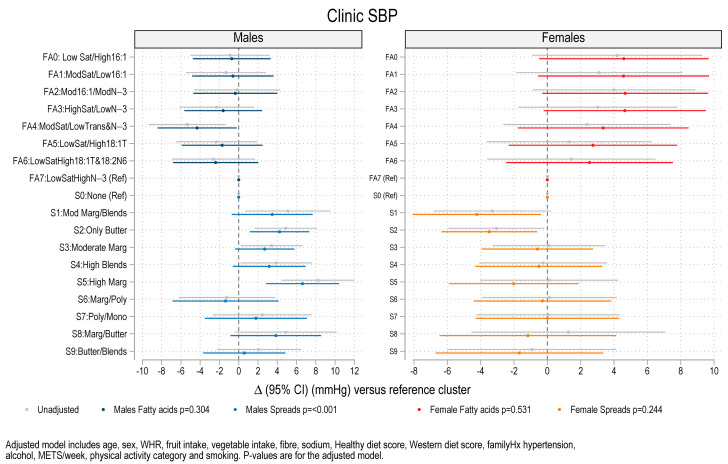
Coefficient plot showing mean difference and 95% confidence interval (CI) in systolic blood pressure (SBP) for each cluster versus reference category clusters (n = 785). Coefficients were estimated from regression models for males and females separately. Legend: FA0–FA7: Fatty acid clusters. S0–S9: Spread clusters. Mod = Moderate, Marg = Margarines, Blends = Butter and margarine blends, Poly = Polyunsaturated margarine, and Mono = Monounsaturated margarine. WHR = waist–hip ratio, Hx = history of, and METs = Metabolic equivalents.

**Figure 4 nutrients-17-00869-f004:**
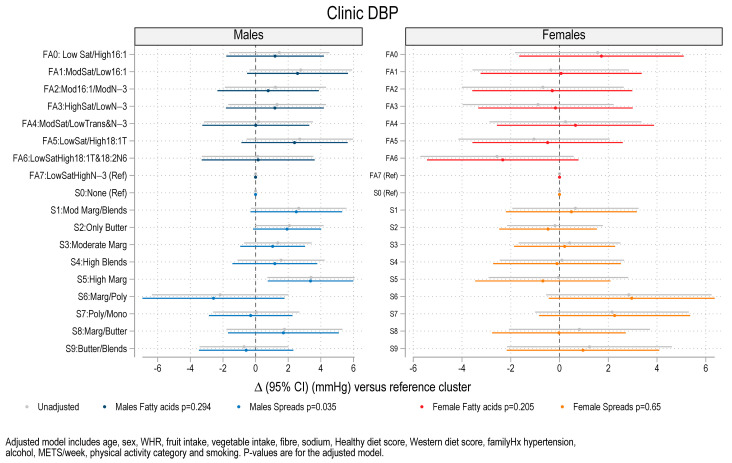
Coefficient plot showing mean difference and 95% confidence interval (CI) in diastolic blood pressure (DBP) for each cluster versus reference category clusters (n = 785). Coefficients were estimated from regression models for males and females separately. Legend: FA0–FA7: Fatty acid clusters. S0–S9: Spread clusters. Mod = Moderate, Marg = Margarines, Blends = Butter and margarine blends, Poly = Polyunsaturated margarine, and Mono = Monounsaturated margarine. WHR = waist–hip ratio, Hx = history of, and METs = metabolic equivalents.

**Figure 5 nutrients-17-00869-f005:**
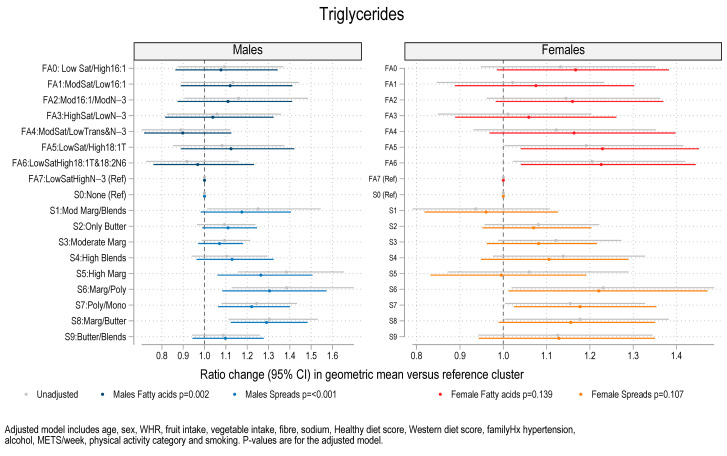
Coefficient plot showing ratio change and 95% confidence interval (CI) in serum triglyceride concentrations geometric mean for each cluster versus reference category clusters (n = 785). Coefficients were estimated from regression models for males and females separately. Legend: FA0–FA7: Fatty acid clusters. S0–S9: Spread clusters. Mod = Moderate, Marg = Margarines, Blends = Butter and margarine blends, Poly = Polyunsaturated margarine, and Mono = Monounsaturated margarine. WHR = waist–hip ratio, Hx = history of, and METs = metabolic equivalents.

**Table 1 nutrients-17-00869-t001:** Characteristics of participants studied at 20 and 22 years (n = 785).

	All (N = 785) (100%)	Males N = 409 (52.1%)	Females N = 376 (47.9%)	*p*-Value ^3^
Age, years At the visit aged 20 At the visit aged 22	19.95 ± 0.42 22.11 ± 0.59	19.99 ± 0.43 22.15 ± 0.62	19.90 ± 0.41 22.06 ± 0.57	0.005 0.041
Body mass index (kg/m^2^) At the visit aged 20 At the visit aged 22	24.2 ± 4.7 25.0 ± 5.0	24.2 ± 4.1 25.2 ± 4.4	24.25 ± 5.3 24.9 ± 5.55	0.960 0.296
Family history ^5^ of high blood pressure, n (%)	244 (31.1)	129 (31.5)	115 (30.6)	0.773
Systolic blood pressure ^1^, mmHg At the visit aged 20 At the visit aged 22	117.2 ± 12.2 118.9 ± 11.3	122.2 ± 11.6 123.4 ± 10.6	111.7 ± 10.3 113.9 ± 9.8	<0.001 <0.001
Diastolic blood pressure ^1^, mmHg At the visit aged 20 At the visit aged 22	65.5 ± 7.3 67.0 ± 7.3	65.1 ± 7.5 66.8 ± 7.4	65.9 ± 7.1 67.3 ± 7.1	0.149 0.328
Stage 1 hypertension ^6^ [34] At the visit aged 20 At the visit aged 22	103 (13.1) 103 (13.1)	74 (18.1) 80 (19.6)	29 (7.7) 23 (6.1)	<0.001 <0.001
Stage 2 hypertension ^6^ [34] At the visit aged 20 At the visit aged 22	27 (3.44) 31 (3.95)	25 (6.1) 26 (6.4)	2 (0.5) 5 (1.3)	<0.001 <0.001
Triglycerides (mmol/L) At the visit aged 20 At the visit aged 22	1.04 ± 0.48 1.06 ± 0.45	1.08 ± 0.63 1.04 ± 0.48	1.13 ± 0.53 1.06 ± 0.45	0.332 0.053
Cholesterol (mmol/L) At the visit aged 20 At the visit aged 22	4.33 ± 0.79 4.62 ± 0.80	4.16 ± 0.74 4.54 ± 0.79	4.52 ± 0.80 4.71 ± 0.81	<0.001 0.003
Non-HDL cholesterol At the visit aged 20 At the visit aged 22	3.00 ± 0.74 3.26 ± 0.77	2.93 ± 0.74 3.30 ± 0.80	3.07 ± 0.73 3.22 ± 0.74	0.010 0.154
HDL cholesterol (mmol/L) At the visit aged 20 At the visit aged 22	1.33 ± 0.32 1.36 ± 0.34	1.23 ± 0.24 1.24 ± 0.24	1.45 ± 0.35 1.49 ± 0.39	<0.001 <0.001
Fruit consumption (age 17), n (%) 0: Rarely or never 1: 1–2 times a month 2: 1–2 times per week 3: 3–5 times per week 4: 6 + times per week Missing	13 (1.7) 33 (4.2) 104 (13.25) 234 (29.8) 252 (32.1) 149 (19.0)	9 (2.7) 21 (6.4) 59 (18.0) 119 (36.3) 120 (36.6) 81 (19.8)	4 (1.3) 12 (3.9) 45 (14.6) 115 (37.3) 132 (42.9) 68 (18.1)	0.247
Fruit consumption category (age 17), Median (IQR) Mean (SD)	3 (3–4) 3.07 ± 0.97	3 (2–4) 2.98 ± 1.03	3 (3–4) 3.17 ± 0.91	0.025 0.014
Vegetable consumption (age 17), n (%) 0: Rarely or never 1: 1–2 times a month 2: 1–2 times per week 3: 3–5 times per week 4: 6 + times per week Missing	1 (0.1) 11 (1.4) 60 (7.6) 212 (27.0) 405 (45.0) 172 (18.85)	1 (0.2) 7 (1.7) 38 (9.3) 114 (27.9) 168 (41.1) 81 (19.8)	0 (0.0) 4 (1.1) 22 (5.85) 98 (26.1) 185 (49.2) 67 (17.8)	0.153
Vegetable consumption cat (age 17) Median (IQR) Mean (SD)	4 (3–4) 3.42 ± 0.75	4 (3–4) 3.34 ± 0.79	4 (3–4) 3.50 ± 0.69	0.012 0.008
Weekly alcohol consumption (age 20) ^2^ Median (IQR) grams per day	8.6 (0–21.4)	10 (0–27.1)	5.7 (0–15.7)	0.132
Alcohol categories (age 20) ^4^ Less than 1 standard drink 1 to 3 standard drinks More than 3 standard drinks	604 (89.5) 47 (7.0) 24 (3.6)	293 (87.7) 26 (7.8) 15 (4.5)	311 (91.2) 21 (6.2) 9 (2.6)	0.287
Smoking status (age 22) No Yes	618 (85.7) 103 (14.3)	311 (83.8) 60 (16.2)	307 (87.7) 43 (12.3)	0.136
Sodium intake (mg/day) (age 17)	2977 ± 1214	3441 ± 1245	2530 ± 999	<0.001
CSIRO diet factor analysis scores (age 17) Western diet score Healthy diet score	−0.06 ± 0.82 0.07 ± 0.90	0.26 ± 0.83 0.01 ± 0.92	−0.36 ± 0.68 0.13 ± 0.89	<0.001 0.163

Data are the mean (SD) or n (percentage). ^1^ Mean of five supine clinic blood pressure readings taken after minutes 2, 4, 6, 8 and 10. ^2^ Using medical questionnaire data at age 20 (available n = 679). ^3^
*p*-value for difference between genders.^4^ One standard drink = 10 grams of alcohol. ^5^ Family history refers to either mother or father. ^6^ Stage 1 hypertension (130–139 systolic or 80–89 mm Hg diastolic) and stage 2 hypertension (≥140 systolic or ≥90 mm Hg diastolic). Data for continuous variables are presented as either the mean ± standard deviation or the median (inter-quartile range). SD = standard deviation. IQR = inter-quartile range. HDL = high-density lipoprotein. CSIRO = Commonwealth Scientific and Industrial Research Organization.

**Table 2 nutrients-17-00869-t002:** Summary of significant differences for fat spread clusters versus the reference cluster (No spreads) amongst males.

	Males	Females
FA0: Low saturated/High 16:1	-	-
FA1: Moderate saturated/Low 16:1	-	Lower HDL (−12.4%)
FA2: Moderate 16:1/Moderate N-3	-	Lower HDL (−10.1%)
FA3: High saturated/Low N-3	-	Lower HDL (−14.0%)
FA4: Moderate saturated/Low Trans and N-3	-	Lower HDL (−15.0%)
FA5: Low saturated/High 18:1T	-	Lower HDL (−16.0%)
FA6: Low saturated/High 18:1T and 18:2N-6	-	-
FA7: Low saturated/High N-3	Reference	Reference
S0: No spreads (Reference)	Reference	Reference
S1: Moderate margarine/blends	-	-
S2: Butter alone	Higher SBP (+4.26 mmHg)	-
S3: Moderate margarine	-	-
S4: High blends	-	-
S5: High margarine	Higher SBP (+6.61 mmHg) Higher DBP (+3.4 mmHg) Higher Trig (+26.4%)	-
S6: Margarine/polyunsaturated	Higher Trig (+30.5%)	-
S7: Polyunsaturated/Monounsaturated	Higher Trig (+22.1%)	-
S8: Margarine/Butter	Higher Trig (+29.1%)	-
S9: Butter/Blend	-	-

S0–S9: Spread clusters. FA0–FA7: Fatty acid clusters. HDL = High-density lipoprotein cholesterol. SBP = Systolic blood pressure. DBP = diastolic blood pressure. Trig = triglycerides.

## Data Availability

Data access is subject to restrictions imposed to protect participant privacy. All researchers using Raine Study data must sign a data access agreement stipulating that data may not be released to anyone other than the investigators of the approved project. Additional details regarding data access are available from https://rainestudy.org.au/ (accessed on 1 December 2024).

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
