# Peer review of "Patterns of Dietary Fatty Acids and Fat Spreads in Relation to Blood Pressure, Lipids and Insulin Resistance in Young Adults: A Repeat Cross-Sectional Study"

_nutrients, 2025, doi:10.3390/nu17050869_

Round 1

Reviewer 1 Report

Comments and Suggestions for Authors

This article, titled "Associations between patterns of dietary fatty acids and fat spreads with blood pressure, lipids and insulin resistance in young adults," investigates the impact of dietary fatty acids and fat spreads on cardiovascular risk factors in young adults. The study employs clustering methodologies to analyze data from the Raine Study Gen2 participants in Western Australia. It provides valuable insights into the associations between dietary patterns and health outcomes, particularly in the context of blood pressure, lipids, and insulin resistance. The research is well-conceived and addresses an important gap in the literature. However, there are several areas that require revision to enhance the clarity and accuracy of the manuscript.

  1. There are format errors in some statements, such as extra punctuation in line 115. and redundant spaces in line 262.
  2. There are errors in the format of references, for example, the year needs to be bold.
  3. The fonts of the professional terms are inconsistent on line 424.
  4. Some sentences have grammatical problems and need to be modified, such as in line 370.

Author Response

Comments 1: There are format errors in some statements, such as extra punctuation in line 115. and redundant spaces in line 262.

 Response 1: We thank the reviewer for their careful attention to the detail of the grammar and punctuation throughout the manuscript. We have re-checked the grammar and punctuation using Microsoft Word's built-in spelling and grammar check and have corrected the following errors that were detected in the manuscript:

New Line 238: Removed redundant comma within "22 and applying".

New Lines 241-243: Added sufficient spacing between words e.g., changing LowSat to Low-Sat.

New Line 253: Removed comma within "22 and applying".

New Lines 247-250: Added spacing in describing Figure 1 legend for fatty acid cluster labels.

New Lines 153-154 and new lines 261-2630. Added commas after each use of "e.g.." to become "e.g.,".

Comments 2: There are errors in the format of references, for example, the year needs to be bold.

Response 2: We have modified references 26, 27 and 43 to bold format for the year of publication.

We have also provided better detail for reference 27 which is a book chapter.

Comments 3: The fonts of the professional terms are inconsistent on line 424.

Response 3: We have now ensured consistent font type (Palatino Linotype size 9 for all references).

Comments 4: Some sentences have grammatical problems and need to be modified, such as in line 370.

Response 4: We have removed commas for the Figure legends of the supplementary materials. Lines 435-438 now reads as:

Figure S5: Coefficient plot showing mean difference (95% CI) in pulse pressure for each cluster versus reference category clusters (n=786). Figure S6: Coefficient plot showing mean difference (95% CI) in cholesterol for each cluster versus reference category clusters (n=786). Figure S7: Coefficient plot showing ratio change (95% CI) in HDL cholesterol geometric means for each cluster versus reference category clusters (n=786).

Reviewer 2 Report

Comments and Suggestions for Authors

The issue of paper is extremly important in our society. Authors presents important data along with their detail analysis.

  1. Abstract: informative, include most important information
  2. Introduction: includes most important information being basis for the presented study results
  3. Methods: well prepared, adequate for aim of studies
  4. Results: presented in form of figures and tables. All results are detail described along with necessary discussion in next section.
  5. Statistical analysis: provided

The paper is suitable for publication in Nutrients but some modifications/improvements are needed. My questions are the following:

  1. The study found notable sex differences in blood pressure and lipid associations. Did you explore potential biological mechanisms or lifestyle factors (e.g., activity levels, overall dietary patterns) that might explain these differences?
  2. Given the cross-sectional nature of the study, how confident are you that the observed associations reflect causal relationships rather than residual confounding or reverse causation?
  3. The study suggests benefits of a low saturated fat, high N-3 diet. How was N-3 intake assessed, and were specific sources (e.g., fish vs. supplements) considered?

Author Response

Comments 1: The study found notable sex differences in blood pressure and lipid associations. Did you explore potential biological mechanisms or lifestyle factors (e.g., activity levels, overall dietary patterns) that might explain these differences?

Response 1: We agree with the reviewer that it would be interesting to further explore the reasons behind the difference between the sexes in the observed associations. Since diet quality and level of physical activity were similar in males and females it is unlikely that these factors could explain the differences in observed associations. We believe a more likely explanation would be the relatively higher overall blood pressures in males and the relatively higher overall HDL-cholesterol levels in females for our study population. The increased heterogeneity in these variables provides greater potential for any cluster-outcome associations to be observed. To acknowledge the observed differences in findings between genders we have therefore now added a paragraph in the discussion section that speculates on the reasons in the observed differences between males and females.

(Lines 370-378). "Our study identified some notable differences in the observed dietary cluster-risk factor associations for the male and female participants, with cluster-blood pressure and cluster-triglyceride associations identified for males but not females and cluster-HDL-cholesterol associations identified for females but not males. One explanation for these different findings might be the different baseline levels for each risk factor between males and females. Males had a an approximate10mmHg higher systolic blood pressure than females at both age 20 and age 22. Similarly, HDL cholesterol levels were approximately 15 to 20 percent higher in females than in males and were more than 40 percent higher in their variability (standard deviations). Triglyceride levels were similar for both males and females at each age, but there were no significant cluster-triglyceride associations for females. However, there was a distinctly similar trend in the observed cluster-triglyceride associations within females to those observed for males".

Comments 2: Given the cross-sectional nature of the study, how confident are you that the observed associations reflect causal relationships rather than residual confounding or reverse causation?

Response 2: We agree with the reviewer that the observed associations may not be causal associations and we were careful to point this out in the discussion section previously (lines 399-400). We also noted that residual confounding is one possible explanation for our findings (original lines 390-395). We did not discuss the possibility for reverse causality but have now added this as a limitation in the discussion section (lines 404-412):

"It is also possible that the observed associations may be due to reverse-causality. For example, participants that had knowledge of having a higher blood pressure or lower HDL cholesterol might influence them to alter their diet to a healthier diet. Participants with normal values may also be tempted to follow a less healthy diet. Such a potential for reverse-causality is however reduced given that the information on both diet and risk factors was recorded within a tight timeframe for each recall of the cohort , and at two separate recalls separated by two years, making the potential for the observed associations to be driven by risk factor knowledge and subsequent dietary changes less likely. The potential for reverse-causality is also less likely given that the direction of the observed associations confirmed previously established causal harmful associations between less healthy diets and higher levels of cardiovascular risk factors".

Comments 3: The study suggests benefits of a low saturated fat, high N-3 diet. How was N-3 intake assessed, and were specific sources (e.g., fish vs. supplements) considered?

Response 3: Section 2.3 describes how the information on N-3 intake was assessed which was via a semiquantitative dietary questionnaire followed by conversion of foods to nutrient intakes via use of the FoodWorks Professional dietary analysis program. In this respect our study did therefore not distinguish between different sources of n-3 fatty acids in the diet. We have therefore added this comment to the discussion section (lines 365-368):

We also did not account for the different sources of the fatty acids obtained from the diet including for example types of fish or the intake of n-3 supplements. However, the impact of the latter would likely be small given that only 28 of 491 participants (5.69%) reported using N-3 supplements at age 17.

Reviewer 3 Report

Comments and Suggestions for Authors
  1. Line 89: the authors refer to the 785-subject sample as “representative” of the Raine Study cohort of Western Australian offspring. On the other hand, only Gen2 participants with complete data for clinic blood pressure, blood lipids, glucose and insulin, and dietary fatty acids at both ages 20 and 22 were included in the current study. Was there any specific protocol to collect these data from a mathematically-selected set of individuals? How can we be sure these participants were representative of the whole cohort?
  2. The computer-based questionnaires mentioned in lines 110-111 should be described in more detail: investigated topics, types and number of items, scale types (numerical, categorical, Likert-similar, etc.), scoring and interpretation, etc.
  3. Please explain abbreviations at first mention in the text and not later – e.g., CSIRO in lines 122 vs 161.
  4. The authors should also introduce in the main text, and not only in Appendix 1, supplementary details on the main fat spreads and dietary fatty acid clustering, as this part of the study protocol is essential for the readers to understand the paper’s fundamental message.

Author Response

Comments1: Line 89: the authors refer to the 785-subject sample as “representative” of the Raine Study cohort of Western Australian offspring. On the other hand, only Gen2 participants with complete data for clinic blood pressure, blood lipids, glucose and insulin, and dietary fatty acids at both ages 20 and 22 were included in the current study. Was there any specific protocol to collect these data from a mathematically-selected set of individuals? How can we be sure these participants were representative of the whole cohort?

 Response1. The selected subjects were based on using a subset of the original participants with complete data rather than being based on any prior specified protocol. As such, we agree with the reviewer that introduces the possibility for selection bias. Since the subjects with complete data might be more adherent to the health study, they may also be healthier than the rest of the Raine cohort subjects that did not have complete data. We have previously alluded to this limitation in the limitations paragraph for the discussion section (lines 397-399: "There may have been selection bias of healthier participants since the number of participants with complete measures used for this 350 study were several hundred fewer than the total number of subjects that responded at ages 20 and 22 to the surveys".  

Comments 2: The computer-based questionnaires mentioned in lines 110-111 should be described in more detail: investigated topics, types and number of items, scale types (numerical, categorical, Likert-similar, etc.), scoring and interpretation, etc.

Response2: We have now expanded section 2 and provided details of the data for each variable collected from the online questionnaires (lines 104-121):

"Waist-to-hip ratio was measured at age 20 and 22 years from a clinical assessment. Computer-based follow-up health questionnaires relating to health and risk-taking behaviours including fruit and vegetable consumption, smoking and drinking was completed at age 20 and 22. Information on whether there was a family history of high blood pressure in either the mother or the father (Yes/No) was recorded at age 8 and was recorded as either "Yes" if either parent had a history and "No" if both parents had no history. Information on level of physical activity at age 20 was recorded as a 3-category variable (low/moderate/high) and as metabolic equivalents (METs) per week obtained from the number of minutes spent per day in moderate activity/vigorous activity/walking. The frequency of fruit consumption and frequency of vegetable consumption were recorded from an online questionnaire at age 17 using a 1-5 Likert scale with categories of rarely/1-2 times per month/1-2 times per week/3-5 times per week/6+times per week. The standard number of alcoholic drinks consumed each day in the past week for each type of 20 different examples was obtained via the medical questionnaire administered at age 20. Responses were converted into grams alcohol per day and categorized as less than 1 standard drink, 1 to 3 standard drinks, or more than 3 standard drinks. Smoking status (yes/no) was obtained via the medical questionnaire completed at age 22. Additional information was obtained from Australia's Commonwealth Scientific and Industrial Research Organization (CSIRO) semi-quantitative dietary recall questionnaire of food records [24,25] that was completed at age 17 including sodium intake (milligrams per day) and, standardized scores for a Healthy diet and for a Western diet that were generated using a factor analysis of major food groups [26]. These two factor scores explained 13 % (Western dietary pattern) and 8.5 % (Healthy dietary pattern) of the total variance in food intake."

Comments 3: Please explain abbreviations at first mention in the text and not later – e.g., CSIRO in lines 122 vs 161.

Response 3: We apologise for not noticing the lack of prior explanation for abbreviations. We have now explained abbreviations for:

CSIRO (new line 117).

FFQ (line 139)

Pulse pressure PP (line 272)

Diastolic BP DBP (line 270)

Comments4: The authors should also introduce in the main text, and not only in Appendix 1, supplementary details on the main fat spreads and dietary fatty acid clustering, as this part of the study protocol is essential for the readers to understand the paper’s fundamental message.

Response 4: We agree with the reviewer and have now added some detail from Appendix 1 into the main text. Lines 167-175 in the main text now read as:

"Briefly, the clustering approach involved 3 steps which used 1) A K nearest neighbors (K-NN) algorithm, 2) Generation of a K-NN network and 3) Community detection. The method has previously been applied to detect clusters of cells and genes [20,21]. For this study, the extracted dietary information on the estimated mean dietary intake of 31 fatty acids estimated at ages 20 and 22 was used with the K-NN algorithm to cluster participants such that those with more similar fatty acid intakes were more likely to be one of the 20 nearest neighbors of one another in a participant-participant matrix. The cosine similarity method was then used to transform the K-NN distances of the 20 nearest neighbors into edge weights for a K-NN network. Finally, the Louvain community detection algorithm was applied to detect the clusters of participants in the network based on their fatty acid intake profiles. The same method was repeated separately for the data on the 5 fat spread types recorded at ages 20 and 22 (12 variables in total including "no spread")."

Reviewer 4 Report

Comments and Suggestions for Authors

Dear Authors,

I am pleased to review your manuscript titled: “Associations between patterns of dietary fatty acids and fat spreads with blood pressure, lipids and insulin resistance in young adults.” The manuscript addresses a highly relevant and timely topic, and I commend you for your thorough investigation. Below are my suggestions and comments:

TITLE

  • I recommend adding the study design to the title for clarity.

INTRODUCTION

  • Line 65-70: Please add a citation to support the statement made in this section.

METHODS

  • In the phrase “per week at age 20..”, the punctuation is incorrect. Please revise it accordingly.
  • The methodology is well-articulated and comprehensive. I commend the authors for the clarity. I would, however, suggest indicating more explicitly that the study adheres to the CONSORT statement.

RESULTS and DISCUSSION

  • In the tables and figures where acronyms are used, please specify the full terms and their definitions in the legends for clarity.
  • The quality of Figures 3, 4, and 5 could be improved to enhance readability and visual appeal.

ADDITIONAL SUGGESTIONS

  • Given the importance and relevance of this study, I would recommend adding a section titled “Implications for Clinical Practice and Directions for Future Research” after the discussion. In this section, it would be valuable to explore the growing role of Artificial Intelligence (AI) in nutrition and health research, particularly in the analysis of dietary patterns and the management of cardiovascular risk factors (doi: 10.3390/nu16132066, doi: 10.1186/s40001-023-01065-y). The integration of AI-based technologies could significantly improve the accuracy and precision of nutritional analyses and contribute to the development of interdisciplinary projects combining nutrition, medicine, technology, and public health (doi: 10.1186/s12909-023-04698-z, doi: 10.1016/j.mex.2023.102525). These advancements could lead to more personalized dietary recommendations and improved monitoring of diet-related health outcomes over time.

Author Response

Comments1 : I recommend adding the study design to the title for clarity.

Response 1: We agree that adding the study design to the title enhances the clarity of the study and have therefore changed the title which now reads as:

Title: Patterns of dietary fatty acids and fat spreads in relation to blood pressure, lipids and insulin resistance in young adults: a repeat cross-sectional study.

INTRODUCTION

Comments 2: Line 65-70: Please add a citation to support the statement made in this section.

Response 2: We have now added the following references to support our statement on the need for more advanced techniques than regression for nutritional epidemiology to account for collinearity.

  1. Goldstein, B.A.; Navar, A.M.; Carter, R.E. Moving beyond regression techniques in cardiovascular risk prediction: applying machine learning to address analytic challenges. European Heart Journal 2016, 38, 1805-1814, doi:10.1093/eurheartj/ehw302.
  2. Bodnar, L.M.; Cartus, A.R.; Kirkpatrick, S.I.; Himes, K.P.; Kennedy, E.H.; Simhan, H.N.; Grobman, W.A.; Duffy, J.Y.; Silver, R.M.; Parry, S. Machine learning as a strategy to account for dietary synergy: an illustration based on dietary intake and adverse pregnancy outcomes. The American journal of clinical nutrition 2020, 111, 1235-1243.

METHODS

Comments 3: In the phrase “per week at age 20..”, the punctuation is incorrect. Please revise it accordingly.

Response 3: We agree with the reviewer and have altered lines 108-109 to read as:

"Information on level of physical activity at age 20 was recorded as a 3-category variable (low/moderate/high) and as metabolic equivalents (METs) per week".

Comments 4: The methodology is well-articulated and comprehensive. I commend the authors for the clarity. I would, however, suggest indicating more explicitly that the study adheres to the CONSORT statement.

Response 4: We have now explicitly referred to the STROBE statement and EQUATOR and have completed a STROBE statement template as supplementary material. Lines 97-101:

The study adhered to the Strengthening The Reporting of Observational Studies in Epidemiology (STROBE) statement guidelines for Enhancing the quality and transparency of health research (EQUATOR) [23]. The selection of these N=785 participants from the full cohort is described in a CONSORT flow chart (Supplementary Figure 1). Informed, written consent was provided by them at each follow-up. A completed STROBE statement for cross-sectional studies is included as Appendix 2.

RESULTS and DISCUSSION

Comments 5: In the tables and figures where acronyms are used, please specify the full terms and their definitions in the legends for clarity.

Response 5: We have now added the full terms for any acronyms used for all Figure legends and Tables.

Comments 6: The quality of Figures 3, 4, and 5 could be improved to enhance readability and visual appeal.

Response 6: We agree that the figures were a little congested and could have used better colours. We have therefore now fully revised figures 3, 4 and 5 as well as the five similar coefficient plots in the supplementary material to improve readability and visual appeal. We hope the reviewer agrees that these plots are much better at conveying the same information.

ADDITIONAL SUGGESTIONS

Comments 7: Given the importance and relevance of this study, I would recommend adding a section titled “Implications for Clinical Practice and Directions for Future Research” after the discussion. In this section, it would be valuable to explore the growing role of Artificial Intelligence (AI) in nutrition and health research, particularly in the analysis of dietary patterns and the management of cardiovascular risk factors (doi: 10.3390/nu16132066, doi: 10.1186/s40001-023-01065-y). The integration of AI-based technologies could significantly improve the accuracy and precision of nutritional analyses and contribute to the development of interdisciplinary projects combining nutrition, medicine, technology, and public health (doi: 10.1186/s12909-023-04698-z, doi: 10.1016/j.mex.2023.102525). These advancements could lead to more personalized dietary recommendations and improved monitoring of diet-related health outcomes over time.

Response 7: We agree with the reviewer for this helpful suggestion and have now included the following section and text (lines 415-423):

Implications for Clinical Practice and Directions for Future Research

Overall, our study highlights the risks with high intakes of margarine or butter and the benefits of a low saturated fat high n-3 diet in a population of young adults suggesting that adherence to a diet with this fatty acid profile should be adopted as early as possible. Our data also suggest that it would be valuable to consider how the growing role of Artificial Intelligence (AI) in nutrition and health research that includes the analysis of dietary patterns such as here and elsewhere [50] might be best utilized to assist in the management of cardiovascular risk factors [51]. The integration of AI-based technologies within healthcare settings has the potential to significantly improve nutritional analyses and contribute to the development of greater interdisciplinary care that combines nutrition, nursing, clinicians, and technology [52]. These advancements could lead to more personalized dietary recommendations [52] and improved monitoring of diet-related health outcomes over time.

Round 2

Reviewer 1 Report

Comments and Suggestions for Authors

No more comments

Reviewer 4 Report

Comments and Suggestions for Authors

The manuscript is of adequate quality. I recommend its acceptance.